# Public investments in the development of GeneXpert molecular diagnostic technology

**Dzintars Gotham**[1]*, **Lindsay McKenna**[2], **Stijn Deborggraeve**[3], **Suraj Madoori**[2], **David Branigan**[2]

**1** Independent Researcher, United Kingdom, **2** Treatment Action Group, New York, NY, United States of America, **3** Médecins Sans Frontières-Access Campaign, Geneva, Switzerland

* dzintarsgotham@gmail.com

## Abstract

### Background

The GeneXpert diagnostic platform from the US based company Cepheid is an automated molecular diagnostic device that performs sample preparation and pathogen detection within a single cartridge-based assay. GeneXpert devices can enable diagnosis at the district level without the need for fully equipped clinical laboratories, are simple to use, and offer rapid results. Due to these characteristics, the platform is now widely used in low- and middle-income countries for diagnosis of diseases such as TB and HIV. Assays for SARS-CoV-2 are also being rolled out. We aimed to quantify public sector investments in the development of the GeneXpert platform and Cepheid's suite of cartridge-based assays.

### Methods

Public funding data were collected from the proprietor company's financial filings, grant databases, review of historical literature concerning key laboratories and researchers, and contacting key public sector entities involved in the technology's development. The value of research and development (R&D) tax credits was estimated based on financial filings.

### Results

Total public investments in the development of the GeneXpert technology were estimated to be $252 million, including >$11 million in funding for work in public laboratories leading to the first commercial product, $56 million in grants from the National Institutes of Health, $73 million from other U.S. government departments, $67 million in R&D tax credits, $38 million in funding from non-profit and philanthropic organizations, and $9.6 million in small business 'springboard' grants.

### Conclusion

The public sector has invested over $250 million in the development of both the underlying technologies and the GeneXpert diagnostic platform and assays, and has made additional investments in rolling out the technology in countries with high burdens of TB. The key role

**Data Availability Statement:** All relevant data are within the manuscript and its Supporting Information files.

**Funding:** This analysis was funded by Treatment Action Group (TAG; https://www.treatmentactiongroup.org/). DG received a research grant for this analysis from Treatment Action Group as an independent researcher. The funder (Treatment Action Group) provided support in the form of salaries for authors DB, LM, SM, but did not have any additional role in the study design, data collection and analysis, decision to publish, or preparation of the manuscript. The specific roles of these authors are articulated in the 'author contributions' section.

**Competing interests:** DG received a research grant for this analysis from Treatment Action Group as an independent researcher. The funder (Treatment Action Group) provided support in the form of salaries for authors DB, LM, SM, but did not have any additional role in the study design, data collection and analysis, decision to publish, or preparation of the manuscript. The specific roles of these authors are articulated in the 'author contributions' section. This does not alter our adherence to PLOS ONE policies on sharing data and materials. SD is an employee of Médecins Sans Frontières. Treatment Action Group and Médecins Sans Frontières have long advocated for lowering the price of GeneXpert tests in low- and middle income countries. The authors declare no other competing interests.

played by the public sector in R&D and roll-out stands in contrast to the lack of public sector ability to secure affordable pricing and maintenance agreements.

## Introduction

The development of molecular diagnostics in the 1990s represented a significant medical advance. Most molecular diagnostics are based on polymerase chain reaction (PCR) technology, allowing the detection of minute amounts of genetic material from a pathogen in a patient sample.

The GeneXpert diagnostic platform is a rapid, automated PCR device that does not require a fully equipped modern laboratory. These characteristics are especially valuable in improving access to molecular diagnostics in healthcare settings where well-equipped laboratories are scarce, or where it is important to provide a diagnosis within hours near the point of care, rather than waiting days to weeks using conventional methods. The GeneXpert diagnostic platform is commercialized by Cepheid, a California-based private company and a subsidiary of Danaher Corporation.

A key example of the importance of rapid molecular diagnostic testing is tuberculosis (TB), the leading infectious cause of death globally, surpassed only in 2020 by COVID-19. Neither of the long-standing diagnostic tests–sputum smear microscopy nor basic culture–can rapidly and accurately diagnose TB. Basic culture is the most accurate TB test and the microbiological reference standard for TB detection, but can take weeks to confirm diagnosis, and sputum smear microscopy is rapid but only about 65% sensitive compared to culture for detecting TB [1]. In comparison, GeneXpert testing can provide accurate detection of TB and rifampicin resistance, with sensitivities compared to culture of 90% and 96% respectively, in under 90 minutes [2]. In 2010, WHO endorsed the Xpert MTB/RIF assay as the initial diagnostic test where there is clinical suspicion of multidrug-resistant (MDR) TB or HIV-associated TB; [3] in 2013, WHO expanded its endorsement of Xpert MTB/RIF as the initial test for all people with signs and symptoms of pulmonary or extrapulmonary TB; [1] in 2017, WHO endorsed Xpert MTB/RIF Ultra, a more sensitive version of the test; [4] in 2020, WHO strengthened its endorsement of Xpert TB tests to be used as initial tests for pulmonary and extrapulmonary TB; [2] and in 2021, WHO endorsed a new Xpert test for resistance to key second-line TB drugs [5]. (The WHO develops these recommendations by convening independent expert groups with declared conflicts of interest, reported in the corresponding policy documents [2, 6]).

Apart from TB, available GeneXpert products include assays for influenza, respiratory syncytial virus, chlamydia and gonorrhea, trichomonas vaginalis, human papillomavirus, group B streptococcus, C difficile, enterovirus, methicillin-resistant Staphylococcus aureus and other drug-resistant bacteria, chronic myeloid leukemia, clotting disorders, HIV, hepatitis B, hepatitis C, and SARS-CoV-2 [7]. Indeed, survey data showed that already in 2014–2016, among 21 high-TB-burden countries, 37% were using the GeneXpert diagnostic platform for other diseases beyond TB [8]. Increasing availability of molecular diagnostics may play an important role in strategies to improve screening and diagnosis for diseases such as human papillomavirus, bacterial meningitis, hepatitis C, and in the midst of the current pandemic, SARS-CoV-2 [9–11]. The ability to test for more than one pathogen at a time is also important to support a differential and definitive diagnosis, such as diagnosing TB versus COVID-19 pneumonia.

Examining public investments can yield insights into the innovation ecosystem as it relates to diagnostics development and downstream access issues, including unaffordable prices for

end users. Where there are substantial public investments in developing a health technology, it is important to examine whether the public sector has received adequate returns for its investments. These goals are, for example, emphasized in the policy governing the licensing of technologies developed by the U.S. National Institutes of Health [12].

We recently investigated the public sector investments in developing bedaquiline, a key new treatment for drug-resistant tuberculosis, and reported that the public sector has invested $455–747 million in the drug's development, estimated to be 1.6–5.1 times the investments made by the proprietor pharmaceutical company [13]. Public investments into early-stage technology development have been key to many modern technologies [14]. While public investments have been reported for the development of pharmaceuticals and vaccines, [13, 15, 16] to our knowledge, no detailed analyses of public investments in the development of a diagnostic technology have previously been published.

We here report the public investments made in the development of the GeneXpert platform and across assays (test cartridges) for a range of pathogens. Our estimates of investment figures cover the suite of assays that can be run on the GeneXpert platform, but we focus our discussion of the findings on TB– the disease area in which the GeneXpert technology has thus far gained the greatest footprint globally, especially in low- and middle-income countries.

## Methods

As public sector investments come from various sources, multiple approaches were used to gather data, outlined below. All dollar values represent U.S. dollars inflation-adjusted to 2020 using the World Bank GDP deflator, [17] unless otherwise indicated. The GeneXpert diagnostic platform (and associated assays, software, etc.) is the only product that Cepheid commercializes; [7] we therefore counted all public sector research funds received by Cepheid as relevant to GeneXpert development.

### Financial reports

Cepheid's annual financial reports (1996–2016), available from the U.S. Securities and Exchange Commission (SEC), were reviewed for references to publicly funded projects, collaboration agreements with public sector entities, research and development tax credits, and relevant intellectual property licensing agreements.

The independent health research and policy think tank, Treatment Action Group has undertaken annual surveys of TB research funders since 2013; research expenditure data from these surveys were reviewed [18]. The non-profit diagnostics developer, FIND (Foundation for Innovative New Diagnostics) returned responses in all years 2013–18; Cepheid did not return responses in 2014 and 2018.

### Public sector grant databases

National Institutes of Health (NIH) funded projects were identified through the NIH Research Portfolio Online Report Tools (RePORTER) database, using the search criteria "Cepheid" or "GeneXpert" in the project description text, and reviewing all NIH grants to one principal investigator (David Alland), as this researcher's laboratory at the University of Medicine and Dentistry of New Jersey is known to have collaborated extensively with Cepheid in developing various GeneXpert assays [19]. Abstracts for projects identified by the search were reviewed by the authors and coded as to whether they pertained to GeneXpert development, GeneXpert validation, or neither (Table 1 in S2 File). Grants were counted as GeneXpert funding if abstracts mentioned GeneXpert or Cepheid as a subject/collaborator, and, for grants to David Alland, where the abstract described work on a point-of-care PCR-based diagnostic (or

mentioned Cepheid or GeneXpert). Grants were considered relevant to GeneXpert pre-approval development when the project focused on developing or improving the GeneXpert device or assay (rather than used the GeneXpert test to address a different research question). Grants were further classed as post-approval 'validation' studies where the grant focused on evaluating the performance of an assay in a real-world setting. Grant databases for other U.S. federal agencies and philanthropic research funders were similarly searched using the terms "Cepheid" and "GeneXpert" (Table 1 in S2 File).

## Literature review

The development history for the GeneXpert technology was described by reviewing materials published by Cepheid and recursively tracing references in these materials backwards in time, as well as through interviews with key informants. Research activities over the past three decades at the Lawrence Livermore National Laboratory (LLNL) and FIND—both identified as key to GeneXpert development through review of Cepheid materials—were investigated further; research activities of key inventors were similarly investigated. A landscape analysis of patents potentially relevant to GeneXpert technology was reviewed for public sector patent applicants and declarations of government funding [20].

## Key informants

LLNL and FIND were contacted with requests for information on expenditures on GeneXpert development. LLNL did not provide information. FIND provided a summary of its investments in projects relating to GeneXpert development.

## Results

### Development history of GeneXpert diagnostic technology

The development history of GeneXpert technology can be seen as consisting of three phases: development of modern PCR techniques, miniaturization and automation of PCR machinery, and development of increasingly sensitive and specialized molecular probes for clinical applications. An overview of the development timeline is shown in Fig 1.

GeneXpert technology is based on the application of microengineering and microfluidics (moving fluids through very small channels) to polymerase chain reaction technology. The use of microfluidics meant that PCR could be done much faster, as smaller amounts of sample-reagent mixture could be heated up and cooled down faster than larger amounts. At the same time, this allowed miniaturization and ruggedization, thereby enabling nucleic acid extraction, amplification, and detection to be performed within a single cartridge-based assay.

The first outline of LLNL's micro-engineered PCR cartridge was published in 1993 [22]. LLNL continued developing this technology, initially focusing on military applications for the detection of biowarfare agents [23–25]. By 1998, LLNL had developed this technology into a field-ready miniaturized PCR analyzer, with funding from the Defense Advanced Research Projects Agency (DARPA) and LLNL institutional funds [26, 27].

Cepheid was founded in 1996 by Allen Northrup, a researcher who had developed these technologies at LLNL, together with others, and Cepheid was granted an exclusive license on relevant patents by LLNL, for a one-time fee of $150,000 paid to LLNL, plus an undisclosed royalty on sales. The technologies developed at LLNL form the basis of Cepheid's commercial products [28].

In the first years after the establishment of Cepheid, further product development depended heavily on defense contracts. Fears of anthrax bio-terrorism after 2001 resulted in Cepheid

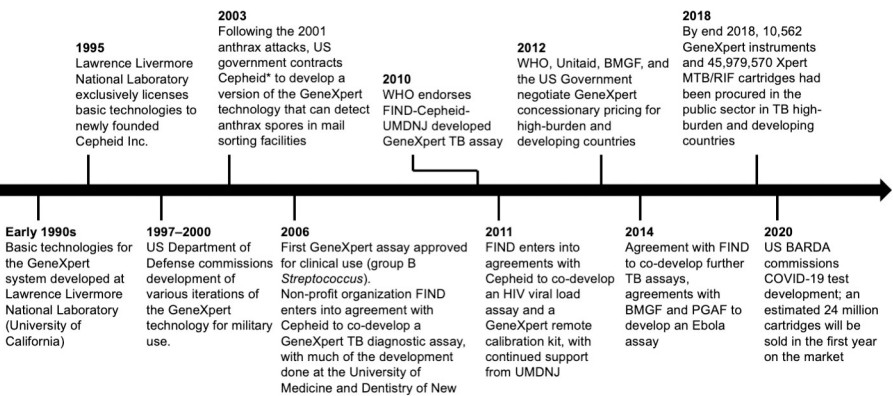

**Fig 1. Timeline of GeneXpert technology development.** *As part of a consortium led by Northrop Grumman. BARDA–Biomedical Advanced Research and Development Authority; FIND–Foundation for Innovative New Diagnostics; BMGF– Bill & Melinda Gates Foundation; PGAF–Paul G. Allen Family Foundation; WHO–World Health Organization. COVID-19 cartridge sales in first year estimated based on announcement that 6 million tests have been sold in one quarter [21].

receiving large contracts to equip the U.S. Postal Service mail sorting facilities with PCR-based anthrax-detection equipment [29, 30].

Up until 2006, Cepheid's GeneXpert technologies were approved for research use only. In 2006, the first GeneXpert assay was approved for clinical use– a group B *Streptococcus* diagnostic (of key importance in obstetrics) [31]. Also in 2006, Cepheid signed a Cooperative Research and Development Agreement with FIND to develop a TB diagnostic that also detects resistance to the first-line TB drug rifampicin—the Xpert MTB/RIF assay—as well as an agreement with the U.S. Centers for Disease Control and Prevention (CDC) to develop influenza diagnostics [32]. Following this, numerous new assays were developed, such as assays for MRSA in 2007, chlamydia and gonorrhea in 2013, HIV viral load in 2014, breast cancer subtypes in 2017, hepatitis C in 2018, and SARS-CoV-2 in 2020.

Over this period Cepheid's GeneXpert platform has been further miniaturized and made portable, with a single-module, battery-operated, computer-independent platform (GeneXpert Edge) launched in 2018, and a similar platform (GeneXpert Omni) adapted for more challenging environments, which is currently undergoing independent evaluation by FIND and is expected to be commercially launched in 2022.

## Public investments in development at the Lawrence Livermore National Laboratory

The key engineering innovations underlying GeneXpert technologies, as well as early versions of the product and field testing, were done at LLNL. However, publicly available information on budgets and expenditures at LLNL does not report granular data on expenditures for specific projects, technologies, or expenditures by specific teams. Budget requests imply an expenditure of around 659 million USD (inflation-adjusted to 2020; S1 File, sheet "LLNL appropriations") for projects relevant to genetic sequencing over 1987–1995 at LLNL and selected other laboratories under the U.S. Department of Energy, but it is difficult to estimate what proportion of this is attributable to GeneXpert-relevant technologies.

Early development of the technologies underlying GeneXpert at LLNL was funded by the microelectromechanical systems (MEMS) program of the DARPA as well as through LLNL's internal budget [23]. The values of some individual grants from DARPA and the U.S. Army

during this time have been reported by Allen Northrup, a key inventor of the technology, totaling $10,674,919 (inflation-adjusted to 2020 USD), [33] and on this basis we have included this value in our summary, as a minimum estimate (Table 1).

## Public funding identified in Cepheid annual reports

Information on public grants, as well as information on tax credits, were identified in Cepheid annual financial reports, available from the U.S. Securities and Exchanges Commission (SEC) (Table 1). We identified a total public investment of about $73 million across various projects mentioned in Cepheid annual reports (some for work done in collaboration with other entities; Table 2 in S2 File). As businesses are not specifically required to disclose government grants or collaborative work with public entities, these filings do not give a comprehensive picture of public sector investments. For the period 1996–2007, Cepheid reported annual summary values for 'government sponsored research' and 'contract revenues' (the latter of which, by the company's definition, includes research grants, among other things; see Table 3 in S2 File). Over this period, Cepheid reports a total of $22,204,000 for government sponsored research and $22,023,000 for contract revenues. Direct comparison of these values to the totals identified by our survey (Table 1) is limited by the fact that many of the investments identified in our survey funded work not only by Cepheid, but by collaborators such as the University of Medicine and Dentistry of New Jersey. Additionally, it is not clear what proportion of 'contract revenues' are from research grants, and the company has not reported on these items after 2007. Due to these limitations and to avoid double-counting with our project-by-project survey (Table 1), we do not include these poorly defined summary figures in our totals of public investment, or in our overview in Table 1 and Fig 1.

R&D tax credits were estimated based on partial information available in Cepheid's annual reports (see S2 File for methodology).

## NIH funding for university research relating to GeneXpert development

A total of 165 NIH grants were identified in the RePORTER database (211 before de-duplication). Of these, 76 (46%) were considered relevant to GeneXpert development, of which 52 concerned pre-market development and 24 were 'validation' projects (S1 File). Cumulative NIH grants to projects identified as part of GeneXpert technology development and validation were $55,810,433 (inflation-adjusted to 2020 USD), excluding SBIR-STTR grants to Cepheid (counted separately). This comprises $42,077,663 in funding of projects directly developing the technology and $13,732,770 for validation projects (Fig 3).

## Grants from other U.S. government departments and small business grants

In general, publicly available data on grants from U.S. government departments other than NIH are limited. For example, the Department of Defense (DoD) only reports research grants made since December 2014 [34].

For small business grants (small business innovation research [SBIR] grants and small business technology transfer [STTR] grants), a searchable database is available and revealed NIH SBIR/STTR funding of $8,242,582, and DoD SBIR/STTR funding of $1,358,428 (inflation-adjusted to 2020 USD) awarded to Cepheid.

## Investments by non-profit and philanthropic organizations

The main non-profit organization that has played a key role in GeneXpert development is FIND, a non-profit established in 2003. From 2007 to the end of 2020, FIND has collaborated

**Table 1. Overview of public contributions to GeneXpert development.**

| Year | Project name | Funder(s) | Amount (USD)[A] |
|---|---|---|---|
| 1992[B]–1998 | Development of underlying technologies at Lawrence Livermore National Laboratory before creation of Cepheid | U.S. Department of Energy, U.S. Department of Defense (DARPA, U.S. Army) | ≥10,674,919[B] |
| 1998–2006 | SBIR/STTR funding | Department of Health and Human Services, U.S. Department of Defense (U.S. Army) | 9,601,010 |
| 2002–2018 | NIH grants for research relating to pre-market technology development | U.S. Department of Health and Human Services (NIH) | 42,077,663 |
| 2011–2020 | NIH grants for validation research (testing real-world effectiveness) | U.S. Department of Health and Human Services (NIH) | 13,732,770 |
| 1996–2020 | R&D tax credits (see S2 File) | U.S. federal and state governments | 66,815,060 |
| 2001[C] | Micro-fluidic Integrated DNA Analysis System (MIDAS) | U.S. Department of Defense (Edgewood Research, Development and Engineering Center) | 3,455,708 |
| 1997 | U.S. Army 'specified device' | U.S. Department of Defense (U.S. Army) | 8,141,691 |
| 1998 | DARPA 'specific device' | U.S. Department of Defense (DARPA) | 6,191,523 |
| 2000 | Soldier Biological Chemical Command project | U.S. Department of Defense (Soldier Biological Chemical Command) | 2,649,719 |
| 2003 | USPS BDS Program–Northrop Grumman consortium–first phase | U.S. Postal Service | 38,837,446[D] |
| 2006 | Xpert MTB/RIF assay development | FIND | 8,936,677 |
| 2006 | CDC influenza POC test | U.S. Department of Health and Human Services (CDC) | 3,857,962 |
| 2011 | Xpert HIV-1 VL assay development | FIND | 5,965,551 |
| 2011 | GeneXpert remote calibration kit | FIND | 1,169,716 |
| 2014 | Xpert MTB/RIF Ultra assay development | FIND | 3,314,496[EF] |
| 2014 | Xpert Ebola assay development | Paul G. Allen Family Foundation and the Bill & Melinda Gates Foundation | 3,756,428[F] |
| 2016 | Finger-stick HIV viral load blood test | Bill & Melinda Gates Foundation | 4,645,887 |
| 2017 | Xpert MTB/XDR assay development | FIND | 2,122,481 |
| 2017 | Finger-stick TB triage blood test | U.S. Department of Defense (U.S. Army) | 3,788,788 |
| 2020 | Xpert SARS-CoV-2 assay development | U.S. Department of Health and Human Services (BARDA) | ≥4,700,000 |
| 2007–2020 | FIND expenses on collaborative projects with Cepheid developing GeneXpert technology, not captured above, including the Omni platform | FIND | 8,045,641[G] |
| | | **Total:** | **252 million** |

Full sources for figures cited in this Table, inflation-adjustment calculations, and other notes are available in the S2 File.

CDC–Centers for Disease Control. FIND–Foundation for Innovative New Diagnostics. BARDA–Biomedical Advanced Research and Development Authority. DARPA–Defense Advanced Research Projects Agency. NIH–National Institutes of Health.

[A] Values have been inflation-adjusted to 2020 U.S. dollars, except where noted otherwise.

[B] LLNL development of technologies directly related to GeneXpert may have begun before 1992, and we consider this figure to be a minimum estimate as there is limited published information on federal grants in this period.

[C] Exact year(s) is not clear; earlier than 2001.

[D] Estimated range $36,409,637–41,265,255.

[E] The value represents the FIND contribution and excludes funding from Rutgers New Jersey Medical School, which is assumed to be covered by the identified NIH grants to this institution.

[F] Reported in the relevant source as funding 'up to' the cited amounts.

[G] Calculated as the difference between FIND investments reported in Cepheid financial filings (all other FIND projects listed in the Table) and total 2007–2020 FIND investments in Cepheid collaborative projects ($29,554,562), as reported by FIND in response to a request sent by the authors. Value not inflation-adjusted.

with Cepheid on eight projects, of which two are ongoing. Over this period, FIND expenses on these projects totaled $6,740,785 for clinical and laboratory studies, $19,688,602 on other

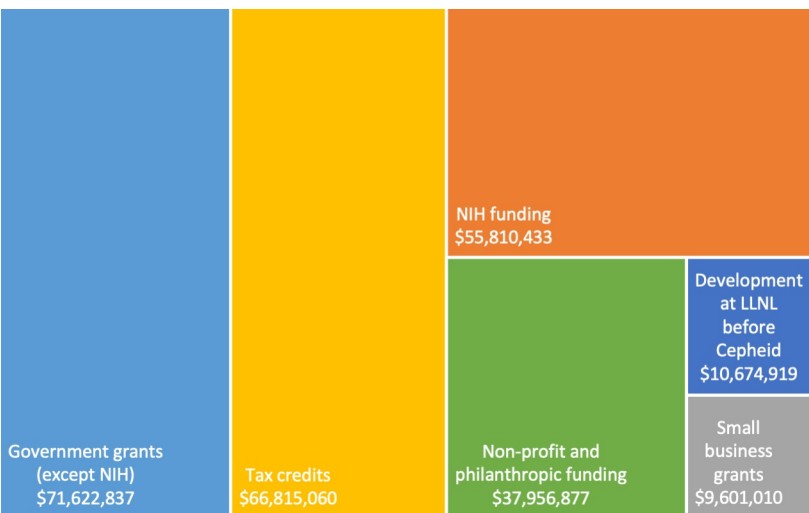

**Fig 2. Summary of public sector investments in GeneXpert development.**

research and development costs, and $3,125,175 on relevant FIND internal operating expenses, such as staff and travel costs. We additionally identified grants by the Bill & Melinda Gates Foundation and the Paul G. Allen Family Foundation (Table 1).

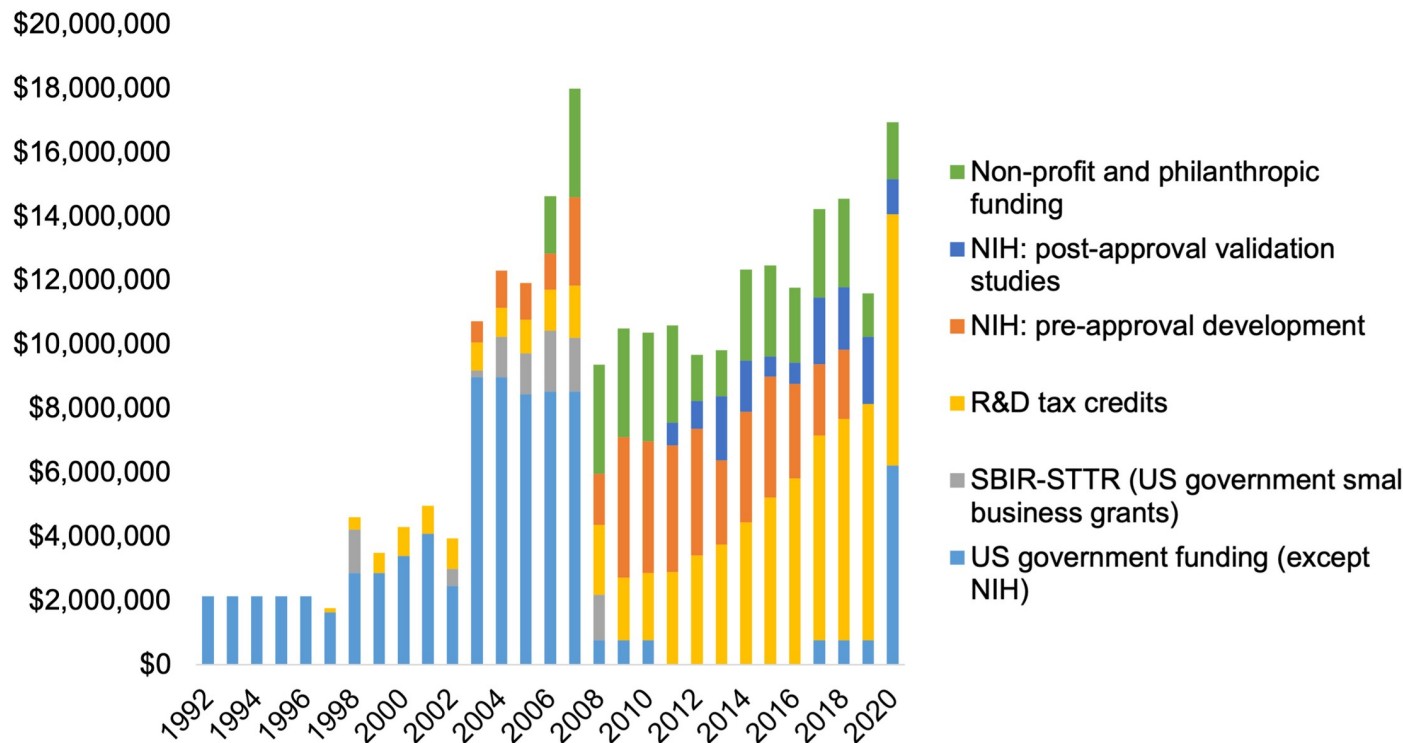

**Fig 3. Public sector investments in GeneXpert development by source and year.** Funding from non-NIH US government departments and non-profit/ philanthropic organizations have been assumed to be equally distributed over a 5-year period to smooth out year-on-year changes.

## Overall public investments

Overall public investments identified in this analysis were $252 million, comprising $55.8 million in NIH funding, $71.6 million in grants from other U.S. government departments, an estimated $66.8 million in R&D tax credits, $38.0 million in funding from non-profit and philanthropic organizations, at least $10.7 million in government-funded technology development before licensing to the private sector, and $9.6 million in small business grants (Fig 2 and Table 1).

Public sector investments have increased over time, with estimated annual investments fluctuating in the range of $8–18 million. Public funding was driven by the U.S. Department of Defense from the early 1990s up to around 2007–2010, when the main public investments became NIH grants, funding from non-profit and philanthropic sources, and R&D tax credits (Fig 3). Nearly all NIH grants (based on analysis of grant abstracts) have been for assay development. Only one grant (4-year time frame, 8.5 million USD) was clearly contributing to device development as well as assay development.

In terms of disease categories, the greatest amount of public investment for GeneXpert assays has been for TB diagnostics, followed by assays for broad multi-bacterial assays (aimed predominantly at screening for agents considered to be a bioterrorism threat, as well as sepsis), with COVID-19, HIV, Ebola, influenza, HPV, cancer, and chlamydia assays also receiving funding, but significantly less than the first two categories (Table 4 in S2 File). However, the majority of public investments identified in this analysis could not be matched to a specific disease category.

## Discussion

Substantial public funding has supported the development of GeneXpert diagnostic technologies. Our survey of the various public investments contributing directly to the development of the technology yielded a total public investment of $252 million. This should be seen as a lower-bound estimate, as certain public investments were likely not captured due to a lack of published information–for example, investments in early development and some military investments.

Public sector funding has come through various routes: military and other federal funding of early-stage development of the underlying microfluidic PCR techniques at a U.S. Department of Energy laboratory; military commissions for product development once the private company was spun off; small-business grants; a commission to develop equipment for the U.S. postal system to detect biowarfare threats; NIH grants for disease-specific assay development; development support from a philanthropically funded non-profit organization; and government tax credits. Overall, U.S. government agencies represent a significant majority of public sector investments in the development of GeneXpert.

Public sector funding can also be seen as having acted as a 'shepherd' for the GeneXpert technology, with the key steps in development triggered by public sector actors' calls for technologies to suit their needs, such as for military field use in the 1990s, anthrax spore detection in a non-laboratory setting, and application to TB initiated by a non-profit organization.

The first phases of GeneXpert development were bolstered by large investments aimed at military and counter-terrorism applications, arguably representing a case study of the tendency for U.S. government funding for global health initiatives to be framed as part of 'national security' [35]. Purchases by the U.S. Government have made up a large part of Cepheid's revenues since the company's establishment, with over $470 million in purchases reported in a government procurement database [36].

Public investment can be highly effective in neglected research areas—diagnostics, especially for infectious diseases, have been highlighted as a historically neglected area within health product R&D [37]. This is being increasingly recognized by research funders, and funding for diagnostics R&D in certain areas, such as for the 'emerging' Ebola and Zika viruses, has seen an increase over the past decade [38]. More recently, BARDA awarded Cepheid an initial grant of $3.7 million to develop a COVID-19 assay, which was granted emergency use authorization less than two months after a public health emergency was declared in the U.S. [39]. Following this, the U.S. government has granted Cepheid at least $54.8 million worth of procurement contracts for the test, illustrating the speed that government-backed diagnostic R&D can take place when there is sufficient political will and guarantees of large purchase volumes.

The development history of GeneXpert also outlines the enormous effect that non-profit initiatives can have on the technology landscape. Recognizing GeneXpert as a potentially revolutionary technology for TB testing, FIND initially approached Cepheid to form a partnership to collaboratively develop Xpert MTB/RIF, a cartridge-based assay for detection of TB and resistance to rifampicin to replace smear microscopy and culture as the initial TB test. In the partnership, Cepheid provided the base technology and the engineering capacity, while FIND provided the technical know-how for the optimization and evaluation of the TB assay. Support from FIND facilitated Cepheid's shift from being a company that primarily manufactured testing platforms, to one that also manufactured assays, assisting the company's expansion into other disease areas. FIND has continued to oversee clinical trials and assay optimization for Xpert TB tests, including the new Xpert MTB/XDR assay for extensively drug-resistant (XDR) TB.

Academic laboratories have supported the development of GeneXpert assays throughout the technology's history, reflected in the $55.8 million in NIH funding (Table 1), of which $34.7 million or 62% was for projects run by David Alland at the University of Medicine and Dentistry of New Jersey (now part of Rutgers University). NIH funded work included, for example, development of assays for MDR-TB, XDR-TB, and Ebola, as well as the development of rapid tests for bacteremia. NIH funded work also included, importantly, numerous 'validation' studies, which test the performance of GeneXpert assays to ensure that they perform as well as (or better than) the earlier standard of care (Fig 3).

PCR-based diagnostic methods are covered by an extensive network of patents held predominantly by large medical device companies. These patents have in many cases led to monopolies [40, 41]. However, many of these methods were developed in the public sector. For example, in addition to the patent that LLNL licensed exclusively to Cepheid, other patents licensed to Cepheid include patents originating from the University of Utah and Baylor College of Medicine [20, 40]. Thermal cycling techniques applied in the GeneXpert platform were licensed from Applied Biosystems, [28] whose technology is in turn based on technologies developed at the California Institute of Technology [42].

The development history of the GeneXpert diagnostic platform has notable parallels to another important tool in fighting the TB epidemic—bedaquiline, the first new drug to be approved for drug-resistant TB in nearly five decades. Bedaquiline similarly benefitted from U.S. military research and extensive public sector investments in its development. An earlier analysis by some of the authors of this study estimated that the public sector invested US$455–747 million in the drug's development, compared to an estimated US$90–240 million invested by the proprietor pharmaceutical company. And, similarly, there have been disputes regarding the fairness of the prices charged by the company [13]. Public investments are especially valuable in areas of health R&D that are historically neglected by the private sector. However, technologies developed through public investments can ultimately be unaffordable to health

systems in the absence of access requirements as a condition of public funding, or if rights are transferred to the private sector without access safeguards and without public accountability for fair pricing [43].

## Pricing and access

In response to early requests for Cepheid to lower prices by civil society organizations, Cepheid in September 2011 published a partial outline of public sector investments in the Xpert MTB/RIF assay, though this outline is no longer available on their website. In that communication, Cepheid estimated that "approximately $37Mil USD has been invested in the development of the Xpert MTB/RIF test by Cepheid, FIND and NIAID", of which Cepheid contributed "in excess of $25Mil USD" [44]. Cepheid's accounting of R&D investments was limited to the Xpert MTB/RIF assay and did not appear to cover investments in the GeneXpert platform more broadly.

In 2012, WHO/Unitaid, the Bill & Melinda Gates Foundation, and the U.S. government (United States Agency for International Development [USAID] and the President's Emergency Plan for AIDS Relief [PEPFAR]) negotiated a 'buy-down' agreement with Cepheid, wherein $11.1 million was paid to Cepheid in return for Cepheid offering the GeneXpert tuberculosis assay cartridge at a ceiling price of $9.98 per test to 145 eligible high-burden and developing countries [45]. This price reduction has been key to expanding use of the GeneXpert platform in these countries, and, in turn, strengthening the evidence base and clinical experience in using the technology [46]. Civil society advocates continue to assert that a further price reduction is warranted given the large annual volumes of tests procured through the public sector and the findings of an independent 'cost-of-goods-sold' (COGS) analysis [47].

While, on the one hand, GeneXpert technology has offered a breakthrough for TB testing and treatment programs, many low- and middle-income countries have been constrained from fully scaling up GeneXpert testing in accordance with WHO recommendations, mainly due to high prices [48]. A standard 4-module GeneXpert machine is priced at $17,000 ('concessionary' price available to certain high-burden countries), with significant maintenance costs (the extended warranty costs an additional ~$2,000–3,000 annually). Cartridges are priced at $9.98 for TB tests (one cartridge is expended for each test of a patient sample) for eligible low- and middle-income countries until 2022, when the current buy-down agreement will end, compared to an average cost of $3.83 for sputum smear microscopy, [49] and up to $19.80 for other pathogens, such as Ebola [50].

Under a 2006 agreement between FIND and Cepheid for development of the Xpert MTB/RIF assay, Cepheid agreed to price cartridges based on the 'cost-of-goods-sold' (COGS) (this includes costs of materials, labor, and overhead) plus intellectual property licensing costs and a 20 percent profit; annual independent audits of these costs; and annual volume- and COGS-based price adjustments [51]. The 'buy-down' price subsidy agreement in 2012 effectively superseded the FIND-Cepheid agreement, and while it set a ceiling price of $9.98, it did not include the volume-based price adjustment clauses that had been included in the 2006 Cepheid-FIND agreement [44]. The volumes projected at the time of the 2012 buy-down agreement (upon which the $9.98 price was determined) were reached already in 2014 and substantially increased over the following years [52]. It can be assumed that these economies-of-scale have substantially lowered the cartridge cost of manufacture. However, the price has remained the same. Thus, the 2012 buy-down agreement lowered the price of TB tests in the short-term but did not put in place long-term conditions for fair pricing.

Independent analyses of the cost of manufacture for GeneXpert cartridges estimate the cost to Cepheid to be $2.95–4.64 per cartridge based on assumed sales volumes of over 10 million

units, a threshold which has been exceeded by public sector procurement for TB testing [47]. Based on these estimates, civil society organizations have called for Cepheid to drop the price to $5 per test [52].

To our knowledge, aside from the 2006 FIND agreement with Cepheid, which was super-seded by the buy-down, no other public funding agreements with Cepheid included terms requiring fair pricing. Arguably, the substantial public investments in GeneXpert's develop-ment have not been reflected in its pricing and the company's commercialization strategy (of maintaining a monopoly). In order for the public sector to receive an adequate return on investment in the development of health technologies, it is in our view essential that public research-funding agreements with the private sector include conditions that require the shar-ing of manufacturing rights and know-how, transparency of the cost of manufacturing or 'cost of goods sold' (COGS), and mechanisms to ensure fair pricing based on COGS with volume-based price adjustments.

## Roll-out and implementation

International global health actors have made substantial investments in expanding access to GeneXpert diagnostics, contributing significantly to the extensive footprint of more than 11,000 GeneXpert devices in use across low- and middle-income countries [53]. Whether or not public investments in roll-out and implementation should be counted as part of 'develop-ment' is up for debate. We have not counted these 'roll-out' and implementation projects as part of our estimate of total public investments in GeneXpert 'development' and did not attempt to produce a comprehensive overview of such projects.

Since the development of the first GeneXpert TB assay, international organizations includ-ing Unitaid, USAID, BMGF, the Global Fund to Fight AIDS, Tuberculosis and Malaria, and others have invested hundreds of millions of U.S. dollars in the procurement of GeneXpert instruments and cartridges. In addition to the 'buy-down' agreement mentioned above, which enables eligible countries to access concessionary pricing, by the end of 2018, a cumulative 10,562 GeneXpert instruments and 45,979,570 Xpert MTB/RIF cartridges had been procured in the public sector in high-TB-burden low- and middle-income countries [54].

It is also worth noting that the U.S. federal government has likely been Cepheid's largest customer in most years of the company's existence. For example, under the U.S. Postal Service Biohazard Detection System program, Cepheid received production contracts for "up to $200 million in anthrax test cartridges and associated materials" in 2007, "up to $112 million of anthrax test cartridges and associated materials" in 2011, and a further contract in 2012 for an undisclosed amount [55]. From the beginning of 2001 to July 2020, the USAspending.gov database lists a total of $476,554,644 in federal procurement contracts awarded to Cepheid (not inflation-adjusted; see S3 File). More recently, the US Department of Defense has awarded Cepheid at least $54.8 million in COVID-19 assay purchase contracts [56–58].

## Limited transparency in R&D investments

Our analysis illustrates the challenges in compiling the development history for a health tech-nology, which are compounded by a lack of availability of both private and public R&D spend-ing. Total public investments in developing GeneXpert technology likely substantially exceeded the investments captured in this analysis, as funding data are in many cases not transparently available (Table 2). For example: LLNL does not publish budgets broken down by project or department; the U.S. Department of Commerce NIST-ATP grants program, which funded projects on the miniaturization of PCR technology, and gave hundreds of mil-lions of dollars in R&D grants to private companies annually between 1991 and the early

2000s, has no public record of awards; [59] the U.S. Department of Defense has ongoing agreements with Cepheid for the development of diagnostics, but these agreements are confidential, with no information available on amounts invested by the government [60]. There is no specific requirement for privately held companies to report on government R&D grants or public sector collaborations; while Cepheid's financial filings did report some information, there is no reason to assume it is comprehensive, and Danaher, a corporation that purchased Cepheid in 2016, has not reported such information since the acquisition. These information gaps make it labor-intensive and, in many cases, impossible to analyze the extent of public investments in health technology development.

There have been growing calls for greater transparency in R&D investments for health technologies. For example, in 2019, a World Health Assembly resolution called for Member States to enhance access to information on clinical trial costs, as well as 'subsidies and incentives' [61]. From the experience of this analysis (Table 2), a few specific recommendations can be made. Health technology companies should be required to publish information on public investments and incentives in the technologies they commercialize. The Lawrence Livermore National Laboratory and other government-funded laboratories should publish data on their 'intramural' project expenditures; data on grants by the NIST-ATP program, which (albeit now defunct) amounted to hundreds of millions of dollars, should be published; investments into health R&D by the US Army (including through DARPA and BARDA) should be made public. Philanthropies and non-profit organizations should publish detailed information on grants and internal R&D expenditures.

## Equitable access to publicly funded technologies

Despite the extensive public sector contributions in developing the GeneXpert diagnostic platform, rights to the technology are held entirely by one private company. Public sector entities have no influence on pricing (beyond negotiating as buyers) or who may manufacture the technology. This is a common pattern for publicly developed diagnostics, treatments, and vaccines; public sector funding underwrites high-risk innovation, but the finished technology is commercialized by a private sector monopolist.

Civil society and academics have long argued that public sector funding for research and technology development should come with safeguards to ensure that access to the end product is shared equitably [62, 63]. This would involve public sector entities retaining the power to exercise key rights to intellectual property protecting the technology, and leveraging these rights in specifying equity-oriented criteria in contracts made with private industry. For example, with such power, public sector entities could stipulate that patents will not be enforced in low- and middle-income countries, thus allowing competitive manufacture and lowering of prices. Where patents are blocking manufacturers from entering the market, compulsory licensing is an important legal option that governments should not hesitate to use in countries facing access challenges.

In the context of diagnostics, manufacturing know-how and other trade secrets may be even more important barriers than patent rights, and, unlike compulsory licenses for patents, there are currently limited legal tools to force the sharing of know-how. In addition to requirements for transparency and fair pricing, public sector research funding should also include conditions for sharing of know-how. This could lead to a broader range of diagnostics manufacturers, especially in low- and middle-income countries, which is an important element in improving access and lowering costs.

In the US, federal law provides so-called 'march-in rights' for federal agencies (such as the NIH), wherein agencies can force the grant of patent licenses to competitors if certain public-

**Table 2. Data availability for sources used in this analysis.**

| Data source | Types of data available | Key data gaps |
|---|---|---|
| U.S. Congressional budget appropriations | Budget requests made to Congress with general descriptions of the area of work funded. | Descriptions are too general to identify or estimate amounts relevant to development of specific technologies. |
| LLNL annual expenditure reports | Not publicly available. | No data available. |
| DARPA expenditure | Not publicly available. | No data available. |
| Cepheid annual financial reports (10-K) | Limited descriptions of some public sector grants/collaborations; Government R&D tax credits up to 2016*; Limited descriptions of development history for key technologies; Limited patent licensing information. | Public sector grants are not reported comprehensively or in detail; Patent licensing information is not comprehensive or detailed. |
| TAG TB R&D investment surveys | Limited data on grants given/received for TB-related research, voluntarily reported by a range of participating research funders, institutions, and private entities. | Cepheid did not submit responses in 2014 or 2018; Often very limited descriptions of grants, making it difficult to connect grants to research outputs. |
| NIH Research Portfolio Online Report Tools (RePORTER) database | Comprehensive data on grants made by the NIH. | No significant data gaps. |
| Small Business Innovation Research (SBIR) and Small Business Technology Transfer (STTR) programs | Comprehensive data on SBIR-STTR grants. | No significant data gaps. |
| U.S. Department of Defense Grant Awards Website. | Grant value, recipient, and limited description. | Appears not to include DARPA or JPEO-CBRND grants. Data only available from 2014. Grant descriptions very limited. |
| Department of Defense. Congressionally Directed Medical Research Programs. | Comprehensive data on grants. | No major data gaps. |
| USASpending.gov: Government Spending Open Data | Data on federal procurement (federal contracts, grants, loans, and other financial assistance awards of more than $25,000). | Descriptions of transactions provided in the public database are extremely short, often not possible to ascertain precise purpose of transaction. |
| Bill & Melinda Gates Foundation | Grant value, recipient, and limited description. | Very limited grant descriptions. Large, general operating grants (e.g. those given to FIND) have no public information on planned spending breakdown. |
| Biomedical Advanced Research and Development Authority (U.S. Department of Health & Human Services) | Grants for developing technologies related to COVID-19. | Other than COVID-19 technologies, grant information is not public. |
| National Institute of Standards and Technology | No searchable grants database. | No searchable grants database. |

*R&D tax credits estimated for the period 2016–2020, see S2 File.

DARPA–Defense Advanced Research Projects Agency.

JPEO-CBRND–Joint Program Executive Office for Chemical, Biological, Radiological and Nuclear Defense.

TAG–Treatment Action Group.

interest criteria are triggered: for example, if "to alleviate health or safety needs which are not reasonably satisfied by the contractor". However, no federal agency has ever used these rights, despite numerous requests to do so from civil society and members of Congress [64].

Recently, Cepheid has offered only a limited a number of its COVID-19 diagnostic cartridges–whose development was funded by the US government (Table 1)–for purchase by the WHO-led Diagnostics Supply Consortium for COVID-19, opting to instead offer the great majority on the open market at non-concessional prices to high-income countries [65]. This limited the ability of many low- and middle-income countries, which invested over the past decade in scaling up GeneXpert testing instruments for other diseases, to adequately test for COVID-19, thereby limiting the effectiveness of their pandemic response.

In view of the large public investments in developing the GeneXpert diagnostic platform and assays, and Cepheid's highly profitable business, [65] we believe that the public should have transparent oversight of the cost of production of GeneXpert products, and that the technology should be made available at fair prices that reflect the cost of production (or cost-of-

goods-sold [COGS]) plus a reasonable profit markup, with volume-based price reductions. Public funders of GeneXpert technology should employ the legal mechanisms at their disposal to hold Cepheid publicly accountable on transparency and fair pricing.

## Limitations

This analysis is, to our knowledge, the only published study to quantify, in detail, public sector investments in the development of a diagnostic technology. The analysis is limited by a lack of granular and transparent data on R&D funding, as outlined in the preceding sub-heading. Additionally, where data is available in a relatively transparent format–e.g. from the NIH RePORTER grant database–individual grants are not linked to the relevant mature technology, meaning that broad-based searches must be undertaken with manual analysis of grant abstracts, a strategy that is likely to miss at least some relevant grants. Similarly, our analysis will not have captured research grants that did not mention a limited set of keywords (Table 1 in S2 File) but may still have been key to GeneXpert development. We were only partially able to break down public investments in terms of R&D stage or disease focus (Fig 2 and S2 File), as information from Cepheid annual reports, US agencies (apart from NIH), philanthropic funders, and FIND was not detailed enough to allow this.

## Conclusion

The origins of the GeneXpert technology are found in work done in public sector laboratories. After technology transfer to a private company, the public sector continued to make large investments in the technology's further development, with overall public sector R&D investments of at least $252 million. Additionally, most of the purchases of GeneXpert products have likely been made by the public sector. The key role played by the public sector in the R&D and roll-out of GeneXpert diagnostics stands in contrast to the lack of public sector ability to secure favorable pricing and terms for maintenance agreements.

## Supporting information

**S1 File. Spreadsheet outlining public sector grants, calculation of R&D tax credits, and LLNL budget appropriations requests.**
(XLSX)

**S2 File. Description of sources for public sector investments data, and methodology for estimating R&D tax credits.**
(DOCX)

**S3 File. Spreadsheet containing data on U.S. government payments to Cepheid and Danaher Corporation (extract from USAspending.gov database).**
(XLSX)

## Acknowledgments

The authors are grateful to FIND for providing R&D expenditure data, as well as Kathleen England, Teri Roberts, Zain Rizvi (Public Citizen), and Mike Frick (TAG) for comments on drafts of the manuscript.

## Author Contributions

**Conceptualization:** Dzintars Gotham, Lindsay McKenna, David Branigan.

**Data curation:** Dzintars Gotham, David Branigan.

**Formal analysis:** Dzintars Gotham, David Branigan.

**Funding acquisition:** Lindsay McKenna.

**Investigation:** Dzintars Gotham.

**Methodology:** Dzintars Gotham, David Branigan.

**Project administration:** Lindsay McKenna, David Branigan.

**Supervision:** Lindsay McKenna, David Branigan.

**Writing – original draft:** Dzintars Gotham.

**Writing – review & editing:** Dzintars Gotham, Lindsay McKenna, Stijn Deborggraeve, Suraj Madoori, David Branigan.

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
