## [Decision Letter · Decision Letter 0]

20 Apr 2021

PONE-D-21-06397

Public investments in the development of GeneXpert molecular diagnostic technology

PLOS ONE

Dear Dr. Gotham,

Thank you for submitting your manuscript to PLOS ONE. After careful consideration, we feel that it has merit but does not fully meet PLOS ONE’s publication criteria as it currently stands. Therefore, we invite you to submit a revised version of the manuscript that addresses the points raised during the review process.

This paper has the potential to have a major impact - but also to arouse controversy, so that it is important to take advantage of the reviewers' helpful comments to strengthen the argument. I would encourage you to respond carefully to all the comments. In some cases it may not be possible to fully respond - for example Reviewer 3, in his/her comments on Table 1 (comment 3 of the main comments) requests a further subdivision of investments which may or may not be feasible, and comment 2 of the supplementary suggestions of Reviewer 3 may also be too difficult/time-consuming for a full response. 

We look forward to receiving your revised manuscript.

Kind regards,

Susan Horton

Academic Editor

PLOS ONE

Journal Requirements:

Thank you for stating the following in the Competing Interests section:

DG received a research grant from TAG as an independent researcher, and DB, LM, and SM are employees of TAG. SD is an employee of Médecins Sans Frontières (MSF). TAG and MSF have long advocated for lowering the price of GeneXpert tests in low- and middle-income countries. The authors declare no other competing interests.

We note that you received funding from a commercial source: Treatment Action Group

Reviewers' comments:

Reviewer's Responses to Questions

**Comments to the Author**

1. Is the manuscript technically sound, and do the data support the conclusions?

Reviewer #1: Yes

Reviewer #2: Yes

Reviewer #3: Yes

2. Has the statistical analysis been performed appropriately and rigorously? 

Reviewer #1: N/A

Reviewer #2: N/A

Reviewer #3: N/A

3. Have the authors made all data underlying the findings in their manuscript fully available?

Reviewer #1: Yes

Reviewer #2: Yes

Reviewer #3: Yes

4. Is the manuscript presented in an intelligible fashion and written in standard English?

Reviewer #1: Yes

Reviewer #2: Yes

Reviewer #3: Yes

5. Review Comments to the Author

Reviewer #1: This study looks at the public and philanthropic/not-for profit contributions to the development of the GeneXpert technology. This is an important compliment to the work that has already been done examining the public contribution to the development of pharmaceuticals. Having made their case that there was a substantial amount of public money invested I would like to read some policy recommendations about going forward, e.g., tying public investment to pricing commitments.

1. The authors make the case about the value of the GeneXpert test for TB and should provide some metrics to back up their statement e.g.., false positive and false negative rates and how do they compare with the metrics from the older more traditional tests? Similarly, how do the costs for the GeneXpert test compare to the costs for the older tests both in high-income countries like the US and in low- and middle-income countries?

2. Did the WHO convene a committee to evaluate (and endorse) the test? If there was a committee did the membership have to declare their conflicts and is that information available?

3. Line 120: Explain what the Treatment Action Group is.

4. On page 11 the authors mention data from SEC filings, but this is not described in the Methods. In addition, did the authors search the financial press, e.g., Financial Times, Wall Street Journal?

5. Line 139: What do the authors mean by "relevant materials"?

6. On lines 223-224, the authors spell out DARPA in full, but that should be done the first time DARPA is mentioned on line 171.

Reviewer #2: It was a pleasure to read this thoroughly research and convincingly argued article. Drawing upon multiple sources of financial data and other documentation to elucidate the development of GeneXpert -- and the vast public resources that went into it --, the authors explain in meticulous detail the complex pathway, funding structures, and array of actors, from which this critically important public health intervention has emerged. The article also represents a novel contribution to the literature in that, although similar studies have been done on drugs and biologics (e.g., https://doi.org/10.1093/jlb/lsz019), to my knowledge no such study has been undertaken with respect to a medical device. I also found the authors' incorporation of key informant feedback into their analysis, to corroborate and triangulate what the documents suggest, to underscore the depth of their approach and importance of their findings. In short, Bravo!

There were, however, a few minor points, which I wanted to highlight for the authors' consideration. Space may pose a challenge to incorporating them all. But to the extent they can do so, I think it will further enhance an already strong article. Specifically,

i) Can the authors clarify (in the body of the manuscript) how the tax credits (p. 13, line 267) were calculated? Tax credits are often overlooked in this kind of analysis; so that others might follow suit in the future, it would be helpful to know exactly how the authors arrived at this figure.

ii) I thought there was room to expand the critique in the discussion. The authors' attention to the buy-down agreements is key, especially given that will expire in 2022. However, such agreements seem like a band-aid solution unable to address the fundamental problem that the public's investments in GenXpert were in no way reflected in the many funding agreements, contracts, etc. that attended and structured its development. I would encourage the authors to make this point explicit in some way, space permitting.

iii) Similar to point ii), I think it would be useful (again, space permitting) to allude to potential policy responses to the problem. There is a fairly extensive literature regarding the lack of use of "march-in" rights related to NIH funding. Can the authors point to that literature and/or develop an argument as to why such rights could or will not work in this case? Another mechanism that has received attention is section 1498 of the Patent Act, which would allow the US govt to override patent rights (see e.g., https://papers.ssrn.com/sol3/papers.cfm?abstract_id=3685413). Patents seem to be in place here. Does 1498 provide a potential solution? Like march-in rights it has seldom been used in recent years. Perhaps that highlights another layer of the problem; namely, that the state, through agencies like NIH, is at ease with the status quo and will not push back against the dominant logics of commercialization. Again, I raise this point only to underscore the importance of the article's findings -- not as a critique. By highlighting the non-use of policy tools at the government's disposal, it may raise additional questions to pursue in the future.

iv) Finally, relating to patents, I am a little unclear about the patent landscape that the authors perform. Can the authors clarify that aspect of their research a little? I think that would be helpful, especially if details regarding policy tools that pertain to patents, are integrated in a revised version of the article.

Reviewer #3: I would like to command the authors for this innovative research idea and execution of the work. The authors’ work provides a comprehensive historical overview and estimated grand total of public investments in the development of the GeneXpert technology and Xpert cartridges. These information are very useful in understanding the role in public investments in facilitating the innovation in technologies (diagnostic) for infectious disease, particularly in those disease (e.g. Tuberculosis) that lack considerable private and overall research and development investments.

This reviewer finds this article can benefit from following revisions in presentation of the authors research methodology and findings.

1) It will be important for authors to provide specifics of data identification and extraction process:

a. What specific criteria was use to include/exclude information used for this work

b. Whether there were ‘duplicates’ of information retrieved from multiple databases and reports and how these were screened/identified, and included as final estimates reported in this study?

c. An overview of types of data/information extracted across all data source (the authors provide some of this information in the text, but it’s very difficult to assess this across multiple databases/sources) – the main idea for this is to show what type of information/data is available from each data source/database and this can help advocate the need for transparency and synchronization of financial information for publicly funded R&D activities

d. If multiple screeners/data extractors were used for the study, how were the discrepancies of data interpretation resolved?

e. For certain data sources, it is possible that total funding/investment amount may be for multiple purposes (i.e. not exclusive to development of the GeneXpert systems/cartridges). Did any of the estimates which the authors included in the manuscript fall in to such criteria? If so, how were the cost/funding estimates apportioned to GeneXpert system/Xpert cartridge development (or was it that because fundings were specific to Cepheid, authors assumed that the full funding amount was for GeneXpert/Xpert cartridge)? If latter, it would be important to specify this decision in the methods section.

2) Following up on the comment made in part c of #1 above, inclusion of an illustration/figure summary or a table to provide an organized overview of all sources of information (e.g. databases, financial reports) would help the readers in understanding the sources of data, types of data available from each source, and which of these public funding are relevant to each of of the main phases of the research and development processes of the GeneXpert and Xpert cartridge technologies [the authors break this process as three categories, but this reviewer would like to suggest four distinct categories: 1. development of modern PCR techniques; 2. miniaturization and automation of PCR machinery (specially for GeneXpert system), development of the cartridges (by cartridge purpose and disease type), and clinical research studies for field applications (this also specify by disease type)]

3) Table 1: The authors have constructed this table with an historical overview in mind and it’s very clear way to present how much investments were made at what time period, by whom. However, this reviewer feels that it would be much more useful to present this information by R&D process (including clinical trials), by product type (GeneXpert system vs. Xpert cartridges), and by purpose of the product’s use (e.g. by disease type). In otherwords, it will be very helpful to know that of 252 million dollar total public investment, what was the distribution of investment into specific R&D activities for each Cepheid GeneXpert ‘product’ that is currently in market?

Following are additional suggestions which the authors can revise/improve their discussion to contextualize the authors’ work in the current debate of transparency in R&D investments and public pricing of GeneXpert systems and cartridges in the high burden, lower income settings.

1) Can the authors comment (or perform additional analyses to estimate) how readers should view/interpret the $252 million dollars in public investment can further justify reduction (or at least continuing the 2012 buy down agreement? The authors provide detailed assessment of current buy down agreement, advocate’s perspectives in further reducing the price (or at least continuing the current agreement), but this reviewer feels it’d be important to find a way to link the authors’ work into this issue.

2) What do the authors think about following issue? The current buy down agreement includes at least 20% profit margin for each unit/cartridge sold/procured (and this does not count potentially larger profit gains for units sold in countries outside of the buy down agreement), which roughly marks 100 million ($2 profit per Xpert MTB/RIF cartridges sold x 50 million so far procured globally) in return on their R&D investment (this does not include any other profit margins made in other global sales – GeneXpert systems, other cartridge types, and sales in non-buy-down agreement countries or in private sectors). How should readers view the public return on investment for the $252 million on R&D of GeneXpert and Xpert cartridges for infectious disease? It would be very interesting to provide some perspectives as to how this public investment in the R&D of GeneXpert should be represented in public pricings of GeneXpert related products, especially in countries with limited financial resources (i.e. simply put, if we were to estimate unit cost estimate for public investment as part of the cost of GeneXpert system and/or Xpert cartridges, what would this unit cost estimate for each cartridge procured in the span of 20 years (contextualized for typical duration of US patents)?)

3) The authors discuss ‘limited transparency in R&D investments in the discussion section, but do not discuss how this transparency can be improved. This reviewer feels that the author’s work can provide some guidance (based on their experience in identifying and extracting data for this study) as to how this transparency issues can be improved, particularly for the public investment.

Along with the authors’ (select co-authors) earlier research work on estimated generic prices for novel drugs for drug-resistant TB, this work has the potential to important information on the magnitude and

6. PLOS authors have the option to publish the peer review history of their article (what does this mean?). If published, this will include your full peer review and any attached files.

Reviewer #1: **Yes: **Joel Lexchin

Reviewer #2: **Yes: **Matthew Herder

Reviewer #3: No

---

## [Author Response · Author response to Decision Letter 0]

5 Jul 2021

Dear Editor and reviewers,

Thank you for your careful review of our submission and thoughtful comments, which have improved the manuscript. Please the attached cover letter for our point-by-point response to your comments. Please note line numbers refer to those in the marked-up document, not the ‘clean’ document.

We have also made the requested changes to formatting.

With sincere thanks again,

Dzintars Gotham

---

## [Decision Letter · Decision Letter 1]

18 Aug 2021

Public investments in the development of GeneXpert molecular diagnostic technology

PONE-D-21-06397R1

Dear Dr. Gotham,

We’re pleased to inform you that your manuscript has been judged scientifically suitable for publication and will be formally accepted for publication once it meets all outstanding technical requirements.

Kind regards,

Susan Horton

Academic Editor

PLOS ONE

Additional Editor Comments (optional):

Reviewers' comments:

Reviewer's Responses to Questions

**Comments to the Author**

1. If the authors have adequately addressed your comments raised in a previous round of review and you feel that this manuscript is now acceptable for publication, you may indicate that here to bypass the “Comments to the Author” section, enter your conflict of interest statement in the “Confidential to Editor” section, and submit your "Accept" recommendation.

Reviewer #1: All comments have been addressed

Reviewer #2: All comments have been addressed

Reviewer #3: All comments have been addressed

2. Is the manuscript technically sound, and do the data support the conclusions?

Reviewer #1: Yes

Reviewer #2: Yes

Reviewer #3: Yes

3. Has the statistical analysis been performed appropriately and rigorously? 

Reviewer #1: Yes

Reviewer #2: Yes

Reviewer #3: Yes

4. Have the authors made all data underlying the findings in their manuscript fully available?

Reviewer #1: Yes

Reviewer #2: Yes

Reviewer #3: Yes

5. Is the manuscript presented in an intelligible fashion and written in standard English?

Reviewer #1: Yes

Reviewer #2: Yes

Reviewer #3: Yes

6. Review Comments to the Author

Reviewer #1: (No Response)

Reviewer #2: The revised article addresses the (minor) concerns I had raised on initial review. In particular, the added detail about this area of research and data regarding tax credits are helpful and underscore the authors' overall thesis. To reiterate, I think this article makes a significant contribution given the growing importance of this diagnostic technology and the substantial amounts of public funding that were allocated during the course of its development. I anticipate that it will have a significant impact in terms of ongoing policy discussions about pricing of publicly developed interventions and its focus on a diagnostic, in contrast to drugs and vaccines, makes it to my knowledge a novel contribution to the literature. I encourage the editors to accept and publish the revised manuscript.

Reviewer #3: No further comments - thank you for addressing most of the comments in your revision (I understand that certain comments may not be possible to fully address in the authors' revision)!

7. PLOS authors have the option to publish the peer review history of their article (what does this mean?). If published, this will include your full peer review and any attached files.

Reviewer #1: **Yes: **Joel Lexchin

Reviewer #2: **Yes: **Matthew Herder

Reviewer #3: **Yes: **Hojoon Sohn

---

## [Editor Report · Acceptance letter]

20 Aug 2021

PONE-D-21-06397R1 

Public investments in the development of GeneXpert molecular diagnostic technology 

Dear Dr. Gotham:

I'm pleased to inform you that your manuscript has been deemed suitable for publication in PLOS ONE. Congratulations! Your manuscript is now with our production department. 

Kind regards, 

on behalf of

Dr. Susan Horton 

Academic Editor

PLOS ONE